# Variability of the Main Economically Valuable Characteristics of *Cyperus esculentus* L. in Various Ecological and Geographical Conditions

**DOI:** 10.3390/plants13020308

**Published:** 2024-01-20

**Authors:** Nina G. Kon’kova, Valentina I. Khoreva, Vitaliy S. Popov, Tamara V. Yakusheva, Leonid L. Malyshev, Alla E. Solovyeva, Tatyana V. Shelenga

**Affiliations:** N.I. Vavilov All-Russian Institute of Plant Genetic Resources (VIR), 42,44, B. Morskaya Street, 190000 Saint-Petersburg, Russia; horeva43@mail.ru (V.I.K.); popovitaly@yandex.ru (V.S.P.); kos-vir@yandex.ru (T.V.Y.); l.malyshev@vir.nw.ru (L.L.M.); alsol64@yandex.ru (A.E.S.); tatianashelenga@yandex.ru (T.V.S.)

**Keywords:** *Cyperus esculentus*, yield, oil content, protein, starch, fatty acid composition

## Abstract

This study includes an assessment of the VIR (Center N.I. Vavilov All-Russian Institute of Plant Genetic Resources) chufa collection, grown in various ecological and geographical conditions of the Russian Federation: “Yekaterininskaya experimental station VIR” in the Tambov region and “Kuban experimental station VIR” in the Krasnodar Region during the years 2020–2021. The main indicators of the economic value of chufa accessions were studied: yield structure and nutritional value (oil, protein, starch, and fatty acid profile). The accessions were grown in regions with different climatic conditions. As a result of the study, the variability of the biochemical and yield characteristics and the correlation between the studied indicators and the factor structure of its variability were established. Of the 20 accessions used in the study, the accessions with the highest protein, starch, oil and unsaturated fatty acid contents were selected, which are the most promising for their use as a raw material to expand the range of regional functional food products, as well as for future breeding efforts in the development of new, promising regional chufa varieties.

## 1. Introduction

*Cyperus esculentus* L. is a perennial plant with numerous tuberous roots on which yellowish-brown tubers measuring up to 3 cm in length and 1 cm in thickness are formed. The flowering stems are 40–50 cm tall, stiff and triangular. The leaves are narrow, measuring 5–10 mm in width and up to 90 cm in length, dark green in color and not pubescent. The plant forms clumps and rarely flowers [1]. In cultivation, it is grown as an annual plant.

Chufa is a moisture-loving plant that requires at least 600–650 mm of annual precipitation for its normal development. In cases of lower precipitation levels, it requires artificial irrigation. Uneven distribution of moisture throughout the vegetation period has a negative impact on its yield. In Spain, chufa is a traditional crop grown in Valencia with the use of irrigation; an area of 546 hectares is allocated for chufa cultivation each year [2]. Chufa is cultivated in some European countries, namely Germany, Poland and Hungary, as well as in some African countries, namely Mali, Benin and the Ivory Coast, as an oilseed and nut-bearing plant. In the Russian Federation, there is currently no industrial production of chufa and it is grown only on small farms [3].

The nodules of this plant are considered as one of the earliest food sources known to mankind. It is established that they were cultivated by the ancient Egyptians, starting from 5000 BC [4] Mohammed et al., 2015. Chufa nodules have valuable nutritional qualities due to their protein—about 15%, starch—about 30%, fat—about 35% and carbohydrate—about 20% content, as well as the minerals, trace elements, vitamins C, A and E, organic acids, alkaloids, steroids, flavonoids and other biologically active substances they contain [5,6,7,8].

There are several strategies to increase the yield of agricultural crops such as chufa: the application of complex fertilizers [9] and selection aimed at obtaining high-yielding regional varieties of chufa that are resistant to the local conditions of reproduction [3,10,11,12].

The most dangerous pests for chufa plantations are wireworms, which are the larvae of the click beetle (family Elateridae). These pests damage the tubers. *Dematophora necatrix* Hartig causes tuberous rot in cultivated *Cyperus esculentus* in eastern Spain. This disease spreads through infected tubers. To overcome this problem, the use of hot water purification of *C. esculentus* tubers has been studied [13].

A study on the effect of extreme temperatures on the viability of chufa tubers showed that when storing chufa tubers at low positive (+4 °C) and negative temperatures (−18 °C and −19.6 °C), germination remains unchanged [3].

An oil with a pleasant nutty aroma is obtained from chufa nodules it is close to olive oil in terms of the fatty acid composition. The oil is noted for its high content of unsaturated fatty acids, comprising approximately 80%. Oleic acid is the predominant fatty acid, accounting for up to 70%, followed by 21% of saturated fatty acids, 67% of monounsaturated fatty acids and 12% of polyunsaturated fatty acids [14,15,16,17,18,19].

Chufa is also widely used in the production of a traditional Spanish beverage called “Horchata de chufa” [20]. Additionally, the production of milk from chufa flavored with extracts from the leaves of various plants has been established [21]. Chufa extracts are added to other beverages to enhance their nutritional value: for example, Kunnu, a popular African drink made from millet, sorghum or corn [22]. Mixtures of chufa milk and soy milk are frequently used [23]. Its use in the confectionery industry for the production of various types of candies and cookies is also common [24].

Chufa can be used as a functional ingredient in meat products to increase the contents of dietary fiber, minerals and vitamins. In addition, its antioxidant properties contribute to the extended shelf life of meat products [25].

Extracts from chufa tubers, stems and leaves exhibit antibacterial and antioxidant activity, making them widely used in medicine [5].

Chufa has a valuable carbohydrate composition, making it suitable for functional nutrition for individuals with diabetes [26,27].

The Vavilov Institute is characterized by the presence of a collection of accessions of various crops, which are studied in different ecological and geographical conditions through a network of experimental stations organized in different soil and climatic zones. The ecological–geographic study of this collection of accessions has been carried out since the founding of the institute. Due to this, it is possible to identify the most “successful” varieties, lines and accessions with highly valuable economic traits (high protein and oil content, starch, etc.) and resistance to environmental stress factors in these soil–geographic conditions. In other words, the formation of regionally adapted varieties for different purposes (food, technical, pharmaceutical, etc.) based on the selected accessions is made possible.

Different cultivational climatic conditions have an influence on the main crop structure indicators and biochemical traits, such as oil and protein content. Under the influence of environmental factors, the adaptability of varieties to the conditions of the environment is considered, i.e., the ability of varieties to produce high yields in various soil and climatic conditions [10,28].

The objective of this study is to monitor various economically valuable traits, such as yield, height, number of nodules per plant, nodule weight per plant, mass of 100 nodules, nodule weight per plot, vegetation period, nodule length, nodule width and nutritional value (protein, starch, oil content, fatty acid composition) of chufa tubers under the contrasting conditions of two regions in Russia (the Tambov region and Krasnodar region), which are most suitable for chufa reproduction. The aim is to identify the most promising chufa accessions with high yield and optimal nutritional qualities for their use as a raw material to expand the range of regional functional food products, as well as for future breeding efforts in the development of new promising regional chufa varieties.

## 2. Results

### 2.1. General Statistics

The results of the general statistics showed that the following points can be noted:All morphological, economical and biochemical parameters are significantly higher in the accessions grown on the Ekaterinino RS;In 2020, chufa plants were taller and more precocious and had a higher oil and starch content and a higher yield per plot; in 2021, they had the highest protein content, as well as the greatest number and weight of nodules from the plant (Table 1)

Among the morphological, economic and biochemical traits of chufa, several groups of traits are distinguished based on the variation magnitude (Table 2).

The duration of the vegetative period is the least variable (ranging from 0 in Ekaterinino research station for both years of the study to 3.4% in Kuban research station in 2020). All biochemical traits (protein, oil and starch content) and plant height show low interpopulation variability. The remaining economic and morphological traits have a medium variation level.

The variability of the fatty acid composition ranges from very low, such as with oleic acid (C18:1), to extremely high, such as with lignoceric acid (C24:0).

Characteristics such as plant height, duration of the vegetation period, and oil and protein content remained relatively stable under different environmental conditions. The average variability was analyzed of the traits that depend on cultivation conditions, including tuber length and width, starch content and the weight of 100 tubers. The remaining traits show high variability depending on cultivation conditions.

Among the fatty acids, low variability depending on the year and location of study is observed in palmitic acid (C16:0) and oleic acid (C18:1), while the content of the other acids shows high and very high variability.

### 2.2. Analysis of Variance (ANOVA)

The results of the ANOVA showed that the height of plants was not affected by either the year or the place of research.

While the number and mass of nodules from one plant depend on the year of study, the mass of 100 nodules depends on the place of study; for other economic and biochemical signs, both the place and the year of study affected the analyzed characteristics.

Of the fifteen fatty acids studied, five have no significant influence on both factors; however, the role of interactions is significant for these fatty acids: palmitic acid (C14:0), palmitoleic acid (C16:1), stearic acid (C18:0), linoleic acid (C18:2), eicosenic acid (C20:1) and behenic acid (C22:0) (Table 3).

### 2.3. Factor Analysis of the Correlation System

In the factor structure of variability according to the complex of morphological, economical and biochemical characteristics, five factors are distinguished, covering 82.4% of the variation (Table 4).

The first factor is related to the variation in the number and weight of nodules from the plant and the starch content. The height of plants correlates with the second factor. The third factor is associated with variations in the protein content and the duration of the growing season. The variability of the oil content is associated with the fourth factor, and the mass of 100 nodules is associated with the fifth.

In the factor structure of the variation of the fatty acid content in chufa oil, six factors are distinguished (Table 5), covering 76.2% of the variability.

The first factor is associated with the variation of palmitic acid (C16:0), oleic acid (C18:1), linoleic acid (C18:2), behenic acid (C22:0) and docosadienoic acid (C22:2). Lauric acid (C12:0), myristic acid (C14:0) and heptadecanoic acid (C17:0) are correlated with the second factor. The third factor varies with the content of palmitoleic acid (C16:1) and lignoceric acid (C24:0). The fourth factor is associated with the variability of the content of vaccenic acid (C18:1 c11) and eicosenic acid (C20:1), with the fifth factor related to stearic acid (C18:0) and arachidic acid (C20:0), and the sixth to linolenic acid (C18:3).

### 2.4. Discriminant Analysis

As a result of the analysis, a set of discriminant functions was obtained, allowing us to distinguish accessions studied at different stations in different years with 100% accuracy. Subsequent canonical analysis identified three axes in the space of which the accessions are differentiated (Table 6, Figure 1).

The duration of the growing season is related to the first axis. In the structure of the second axis, the main role is played by the starch content and the number of nodules, and in the structure of the third the length and width of the nodule are most important.

The discriminant analysis model included the most significant parameters: the duration of the growing season (Root 1), the starch content and the number of nodules per plant (Root 2) and the length and width of the nodule (Root 3) (Table 6).

Under the influence of the identified factors, chufa accessions from different locations and years of reproduction were divided. The chufa accessions grown in the Krasnodar region in 2021 and the chufa accessions grown in the Tambov region in 2021 formed separate groups, while the chufa accessions from the 2020 reproduction year in both regions were combined into one conglomerate. The accessions from the Tambov region formed more compact groups, indicating that the chufa accessions grown in the Krasnodar region had a wider range of variability in their characteristics (Figure 1).

The accessions grown at the Ekaterinino research station in 2021 are differentiated along the first canonical axis, while the accessions from the Kuban research station in the same year are differentiated along the second axis. The third axis allows for differentiation of the accessions from the Ekaterinino and Kuban research stations grown in 2020.

Therefore, the reproduction conditions of the chufa accessions in the two selected locations were similar in 2020, which resulted in obtaining practically identical results in field and biochemical studies. The weather conditions in 2021 greatly differed from those in 2020, both overall and in each of the chufa reproduction regions (KRS, ERS), which subsequently affected the results of the field and biochemical analyses.

## 3. Discussion

The chufa strains from the Vavilov Institute Collection have been previously studied. In 2009, an ecological and geographical study was conducted in four geographic locations, including the Kuban and Ekaterinino research stations [11]. In the tubers of the 2009 Kuban research station reproductions, the protein values were higher (5.95%) than the ones we present (5.5%), but the indicator was more variable. The standard deviation from the mean was 1.4 in 2009, while in the current study, it was 0.07. Thus, in 2020–2021, the protein content indicator was more stable. The oil content in 2006 was slightly lower, and the indicator of relative variability was slightly higher (21.73% ± 1.0) compared to the data from 2020–2021 (23.4 ± 0.37). The starch values in 2009, on the other hand, were slightly higher and less variable (22.9 ± 0.77) compared to 2020–2021 (21.725 ± 2.3). In the fatty acid composition of chufa tubers in 2009 and 2020–2021, oleic acid was the dominant fatty acid. The amount of saturated fatty acids decreased from 23.14 to 18.85% due to the levels of palmitic acid (20.025 ± 1.3 in 2006 and 13.631 ± 0.284 in 2020–2021). On the contrary, the sum of the unsaturated fatty acids increased from 76.87 to 81.95% due to the oleic and linoleic acids (63.775 ± 1.9 in 2009 and 65.162 ± 0.738 in 2020–2021; 11.8 ± 1.1 and 14.744 ± 0.379, respectively). The variability of the fatty acid indicators was more stable compared to 2006, according to the indicator of relative variability.

When comparing the reproduction indicators of the Ekaterinino research station in different years, it was found that the protein content in 2009 was higher (7.13 ± 0.61), while the oil and starch content were lower (16.57 ± 1.59 and 14.50 ± 1.56, respectively) compared to the 2020–2021 years (5.9 ± 0.11; 25.6 ± 0.37 and 25.6 ± 0.27%, respectively). At the same time, the indicators for 2020–2021 are more stable. Regarding the fatty acid composition of chufa oil, the reproductions of different years from the Ekaterinino research station (ERS) observed the same trend as discussed above for the Kuban research station (KRS) reproductions. The sum of saturated fatty acids decreased, while the sum of unsaturated fatty acids increased (23.14 and 14.405; 76.87 and 85.564%, respectively). The content of palmitic, stearic and linolenic acids decreased from 20.87 ± 1.82 to 11.818 ± 0.233, from 2.27 ± 0.81 to 1.993 ± 0.163 and from 1.87 ± 0.81 to 0.437 ± 0.137%, respectively. On the other hand, the levels of oleic acid increased from 61.17 ± 1.32 to 69.841 ± 0.575%, while linoleic acid remained almost unchanged (13.83 ± 1.32 and 13.598 ± 0.40%).

In 2009, the protein, oil, starch and unsaturated fatty acid indicators were higher at the Ekaterinino research station, while the saturated fatty acids were higher at the Kuban research station. In 2020–2021, the trend changed, with the protein indicators being higher in the reproductions from the ERS, while the oil and starch indicators were higher in the KRS reproductions. The sums of the saturated and unsaturated fatty acids were almost equal.

In 2004, tubers reproduced in the conditions of the Leningrad region (Pushkin laboratories VIR) demonstrated a lower oil content (13.5 ± 2.4) and starch content (21.6 ± 2.0) and a higher protein content (6.1 ± 1.4) with a higher dispersion of indicators compared to 2020–2021 [12]. The quantity of SFAs (saturated fatty acids) (22%) was higher, while UFAs (unsaturated fatty acids) (78%) were lower compared to the values of 2020–2021. The increase in SFAs was due to the proportion of palmitic acid (20.4%), while the decrease in UFAs was due to oleic acid (57.7%).

Previously, in 2010–2012, a chufa collection was studied in the conditions of the Krasnodar region [3]. The investigation of the economically valuable characteristics of the chufa collection accessions showed the ranges of variability of plant height (from 49 to 69 cm), tuber weight per plot (1 m^2^) (from 101 to 393 g) and oil content (from 13.1 in to 21.06%). Protein content variation (from 6 to 10%) was much wider in 2010–2012 compared with the current investigation data. The oil and fatty acid composition in 2010–2012 was characterized by a higher content of oleic acid, up to 71.29%. The ranges of variability of the content of saturated and unsaturated fatty acids (from 17.75 to 20.99% and 78.15% to 83.72%, respectively) in 2010–2012 were narrower compared with the indicators received in 2020–2021. It should be noted that the saturated fatty acid variability range in 2020–2021 moved to lower values [3].

Thus, the need for constant monitoring of the key indicators of economic value is confirmed under various eco-geographical conditions, especially in light of global climate change [29].

According to Ayeni et al., field research was conducted on tubers reproduced from accessions acquired from New Jersey and New York in 2008–2021. Several economically valuable traits were determined. The plant heights recorded by Ayeni et al. ranged from 91 to 111 cm, which was higher than the values observed in our study. The minimum and maximum number of tubers in our research were higher compared to the data gathered by Ayeni et al. (28–100) [30]. Conversely, the individual tuber weight in our study was significantly lower compared to the data from Ayeni et al. (68–78.6 g) [30]. The oil content reported by Ayeni et al. (20.0–26.4%) was comparable to our findings, while the protein content was higher (6.4–8.1%) [30].

Bertrand Matthäus et al. conducted a study on chufa tubers collected from several regions of Turkey in 2007. The oil content was found to be 17.3%, which is lower than our data. The oil and fatty acid composition also showed some differences compared to our study. The contents of palmitic acid (14.5%) and oleic acid (62.4%) were slightly lower. On the other hand, the contents of linoleic acid (17.0%), linolenic acid (0.6%) and eicosenoic acid (0.6%) were higher, while the contents of stearic acid (2.7%) and palmitoleic acid (0.1%) practically coincided with our data [31].

While making comparisons between fatty acid profiles, significant differences among the plants grown in diverse geographical locations have been found. Such differences could be because of variations in the age of plants, soil conditions, soil nutrient status, soil water status and other unidentified factors which might contribute to such differences. Therefore, differences are bound to exist. But, whether such differences are due to the genotype, due to the environmental conditions or both is not known.

Many authors highlight the significance of chufa as a raw material for obtaining high-quality vegetable oil [31].

Adding chufa flour improves the quality of bread and its fatty acid composition, enhances its nutritional value and reduces the cost of bread production in regions favorable for its cultivation [32]. It is also emphasized that chufa food products can be used to enrich the diet of patients with nutrition-related diseases [32].

When studying the influence of the origin on the biochemical indicators of chufa, it was shown that in the Polinyà region, Valencia province, Spain, the protein content was only 3.3%, while in various regions of Burkina Faso, the fluctuations ranged from 5.6% to 7.3% and in Niger, it was measured at 8.5%. At the same time, in our study, the protein content was 5.5–5.9%, which is close to the data obtained in Burkina Faso [6].

The oil content in chufa grown in the Polinyà region, Valencia province, Spain was 28.2%. In different regions of Burkina Faso, oil fluctuations ranged from 25.4% to 25.8%, while in Niger, its content was reported to be 35.2% [6]. The protein content in chufa grown in Egypt was 4.7% [15].

The oil content in Egypt accessions was 23.9% [15]. In our study, the oil content ranged from 23.4% to 25.6%, which also corresponds to the oil content in Jerusalem artichoke grown in different regions of Burkina Faso and Egypt.

The starch content in chufa grown in Xinjiang, China was 20.5% [33]. In the study by Sandhu et al., it is noted that the structure of chufa starch is similar to corn starch [34]. In our study, the starch content ranged from 22.9% to 25.6%.

## 4. Materials and Methods

### 4.1. Research Materials

Twenty collection accessions of various origins served as the materials for the study (Table 7).

### 4.2. Reproduction Conditions 2020–2021

The field study was conducted in 2020–2021 in the conditions of the Krasnodar region branch “VIR Kuban Research Station” (45°13’ N, 40°47’ E) and the Tambov region branch “VIR Ekaterinino Research Station” (52°98’ N, 40°80’ E), Russian Federation. The preparatory work utilized the methodological instructions for studying the world collection of oil crops edited by Davidyan G.G. [35].

The Tambov region is located in the southern part of the East European Plain and is part of the Central Chernozem region of the Russian Federation. The topology of the region is a lowland plain with a predominant height of 150 m above sea level. The soils in the region are mainly chernozems, rich in humus. The climate is characterized by sharp continental conditions. It has relatively warm summers and long, cold winters. The average air temperature in the warmest month, July, ranges from +19 to +20.7 °C. The Tambov region is classified as an area with insufficient moisture. The annual precipitation amount is 500–550 mm in the north and about 425–475 mm in the south of the region. The sum of the precipitation during the growing season is 50–60% of the annual amount.

The Kuban Experimental Station of the All-Union Institute of Plant Industry is located in the flat eastern steppe zone of the Krasnodar Region, between the cities of Armavir and Kropotkin (65 m above sea level). The soil represents a variety of Priazov chernozem with a thick humus horizon (130–140 cm). The climate of the area is mild and moderately continental, with unstable moisture. On average, over 30 years, the annual temperature is 10.4 °C; in the coldest period (January), it reaches −3.5 °C, and in the warmest (July), temperatures can reach 23.2 °C. In the Krasnodar region, the annual precipitation is approximately 700–750 mm. The maximum amount of rainfall is observed in June and in November–December, while the minimum occurs in August. The average annual precipitation is around 735 mm.

During the research in 2020, both stations experienced a prolonged drought period (July–September at the Ekaterinino RS and August–October at the Kuban RS). In 2021, the drought period was weakly expressed, occurring in August at the Ekaterinino RS and July at the Kuban RS. The Kuban RS also experienced slightly dry conditions in April 2020 and periods of excessive moisture in May 2020 and August–September 2021. The weather conditions, including air temperature and precipitation, on the Ekateryninno RS and Kuban RS during the research period in 2020–2021 were below the average long-term indicators (Figure 2). Comparing the weather conditions at the Ekateryninno RS and Kuban RS for the years 2020–2021, it can be observed that the difference between the average values of the air temperature and precipitation during the vegetation period significantly increased in 2021 compared to 2020 (Figure 2).

### 4.3. Assessment of Useful Agronomic Characters

The field study of each chufa accession was performed in the context of its useful agronomic characters: the plant height, the number of nodules per plant, the weight of nodules per plant, the mass of 100 nodules, the weight of nodules per plot, vegetation period, the length of the nodule and the width of the nodule. The study was carried out in accordance with the methodological instructions for studying the world collection of oil crops edited by Davidyan G.G. [35]. All the chufa accessions were collected for analysis at the stage of full ripeness (Figure 3).

### 4.4. Biochemical Analysis

Each chufa accession was represented by fifteen plants, and measurements were taken from approximately 450 g of mixed nodules. Prior to the analysis, the nodules of each chufa accession were dried at room temperature, then the material was ground into flour using a CM 290 Cemotec laboratory disc mill (FOSS, Sweden). The biochemical analysis was performed at the VIR Department of Biochemistry and Molecular Biology, applying VIR’s methods edited by Ermakov A.I. [36]. Protein, oil and starch content measurements were performed in two replications, and the fatty acid profile in three replications; the obtained average values were statistically analyzed. Values were expressed in “% dry weight”. Protein content was measured by the Kjeldahl method: 300 mg of the milled material was mineralized with 5 mL of concentrated sulfuric acid at 420 °C for 1.5 h. Nitrogen was determined using a Kjeltec 2200 semi-automatic analyzer (FOSS, Hilleroed, Denmark) with an automatic distillation unit, followed by titration with a 0.1 N sulfuric acid solution. The total protein content was calculated from the nitrogen content using a coefficient of 5.5. Oil content was calculated by weighing the dry fat-free residue. The analysis was carried out in a Soxhlet apparatus with petroleum ether used as the solvent (boiling point 40–70 °C). Starch content was measured by the Ewers polarimetric method. Two grams of the material was hydrolyzed in 25 mL of 1% hydrochloric acid solution (in 50 mL volumetric flasks) in a boiling water bath (Lauda Hydro H 20 SW) for 15 min and cooled to room temperature; then, 2.5 mL of phosphotungstic acid was added to precipitate polysaccharides, proteins, etc., and distilled water was added to a volume of 250 mL. The extract, after filtering, was poured into a 10 cm long polarizing cuvette and the rotation angle was measured on an SAC-I automatic polarimeter/saccharimeter (Saitama, Japan). The conversion coefficient was 203.

The fifteen fatty acids in chufa nodule oil were detected by gas chromatography. One hundred milligrams of the milled accession powder were mixed with 1.5 mL of n-hexane, shaken periodically over 20 min, then centrifuged at 10,000× *g* for three min. The hexane fraction was evaporated to complete dryness in flowing nitrogen, then 0.5 mL of a 0.1 Mol solution of sodium hydroxide in methanol was added, and the mixture was heated for 15 min at 100 °C for obtaining fatty acid methyl esters. After cooling, 0.5 mL of n-hexane was put in the tube, and it was shaken vigorously; then, the hexane fraction was transferred into the GC vial [37].

### 4.5. GC-MS Analysis and Fatty Acid Profile Data Processing

The GC analysis was carried out on an Agilent 6850 gas chromatograph with an Agilent 5975B VL quadrupole mass selective detector MSD (Agilent Technologies, Santa Clara, Calif., USA). Fatty acid methyl esters were separated on a Omegawax TM 250 polar column (polyethylene glycol, 30.0 m, 250.00 μm, 0.25 μm; Missouri, USA) under the following heating conditions: the initial temperature was 170 °C, which was kept for 2 min, then the temperature was increased to 220 °C at a rate of 3 °C/min and kept at 220 °C for 5 min; the injection volume was 1.2 μL, and the helium flow rate was 1.5 mL/min.

Fatty acid identification was carried out using the retention time of a standard mixture of fatty acid methyl esters (37 components, 47885U; Supelco, Bellefonte, PA, USA). Chromatograms were processed using the UniChromTM (2010, “New Analytical Systems”, Belarus, www.unichrom.com, access date 18.08.2022) programs. The content of fatty acid methyl esters was calculated by the method of internal normalization; the content of each fatty acid was expressed as a percent of the total fatty acid content.

### 4.6. Statistical Analysis

Statistical processing was carried out using the Statistica 12.0 application software package (StatSoft, Inc. (2019), STATISTICA (data analysis software system), version 12. www.statsoft.com) and included:Calculation of the basic descriptive statistics for the studied accessions (mean, error of mean and coefficient of variation); calculated averages were used for further analysis;Analysis of variance in order to assess the significance of the influence of factors of the place and year of study on the values of the parameters;Factor analysis of the correlation system of the characters;Discriminant analysis in order to select the most informative characters for distinguishing populations grown in different geographical locations in different years.

## 5. Conclusions

It can be concluded that there are few articles dedicated to studying the economically valuable characteristics of chufa tubers, making the current study relevant.

The ecological–geographical study of economically significant crops is important for the development of regional high-yielding varieties with superior nutritional quality. Furthermore, it helps identify accessions with an optimal biochemical composition for different food uses.

The results of studying chufa in two contrasting ecological–geographic conditions showed that the collection accessions demonstrate significant diversity in terms of major economically valuable characteristics and the tubers biochemical composition. Oil extracted from chufa accessions grown in the Tambov region (ERS) accumulated higher amounts of unsaturated fatty acids, mainly due to an increase in the proportion of oleic acid. Accessions: VIR-20 (Ukraine) with an oleic acid content of over 70%; VIR-7 (Poland), VIR-9 (Bulgaria), VIR-20 (Ukraine), VIR-21 (Belarus), VIR-24 (Ukraine), VIR-27 (Russia) with an unsaturated fatty acid content exceeding 85%, which is considered optimal for food consumption, including vegetable oil production.

The following chufa accessions are selected based on a combination of economically valuable characteristics as the most suitable for food consumption: VIR-20 (Ukraine), with high protein, starch and oleic acid content; VIR-27 (Russia), with protein, oil, starch and unsaturated fatty acid content. On the other hand, VIR-14 (Côte d’Ivoire), with high protein, oil and starch content; VIR-12 (Benin) and VIR-16 (Côte d’Ivoire), with high oil and starch content; and VIR-7 (Poland), VIR-8 (Bulgaria) and VIR-9 (Bulgaria), with protein and starch content, are suitable for both food purposes and animal feeding. Furthermore, these selected accessions can be used in breeding as source material for the development of new high-yielding varieties with improved biochemical composition.

## Figures and Tables

**Figure 1 plants-13-00308-f001:**
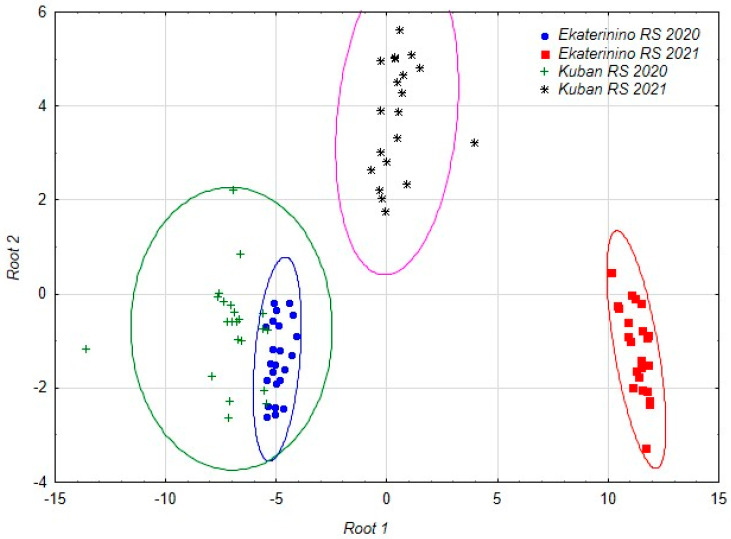
The distribution of the accessions studied in different years at different stations in the space of the first two canonical axes.

**Figure 2 plants-13-00308-f002:**
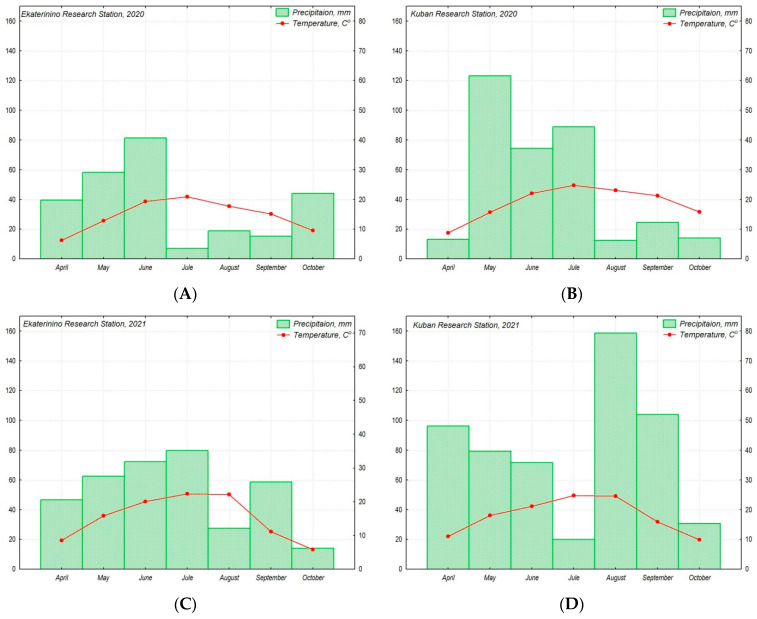
The figure illustrates the climatic conditions of the production of chufa accessions from the VIR collection for the years 2020 (**A**,**B**) and 2021 (**C**,**D**) at the Ekateryninno RS (**A**,**C**) and Kuban RS (**B**,**D**). The amount of precipitation is represented by a bar graph (**A**–**D**) and expressed in mm, while the air temperature is represented by a curve and expressed in °C. The climatic parameters are presented as monthly average values.

**Figure 3 plants-13-00308-f003:**
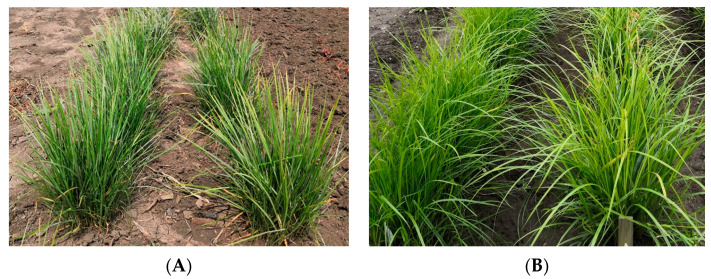
Chufa grown in the conditions of the Kuban RS (**A**) and Ekaterinino RS (**B**).

**Table 1 plants-13-00308-t001:** The magnitude of the characteristics of chufa in various years of research on the Ekaterinino and Kuban research stations.

Character	2020	2021	ERS	KRS	ERS, 2020	ERS, 2021	KRS, 2020	KRS, 2021
H	54.1 ± 0.78	53.6 ± 0.81	54.2 ± 0.74	53.5 ± 0.85	53.7 ± 1.05	54.7 ± 1.05	54.5 ± 1.18	52.5 ± 1.22
NNOD	61.0 ± 2.93	122.2 ± 6.99	92.6 ± 5.42	90.7 ± 8.34	70.5 ± 3.57	114.7 ± 7.95	51.6 ± 3.76	129.8 ± 11.46
MNOD	29.3 ± 1.75	52.6 ± 2.86	43.4 ± 2.13	38.5 ± 3.53	36.2 ± 1.56	50.5 ± 3.40	22.3 ± 2.39	54.7 ± 4.63
NOD100	44.8 ± 1.57	44.3 ± 1.17	46.5 ± 1.39	42.5 ± 1.31	48.3 ± 2.24	44.7 ± 1.59	41.2 ± 1.97	43.9 ± 1.74
MPLOT	373.7 ± 19.43	178.6 ± 12.66	306.1 ± 24.86	246.1 ± 17.39	435.8 ± 26.43	176.4 ± 17.38	311.5 ± 22.24	180.7 ± 18.79
VP	101.2 ± 0.45	122.2 ± 1.47	117.5 ± 2.16	105.9 ± 1.06	103.0 ± 0.00	132.0 ± 0.00	99.3 ± 0.71	112.4 ± 0.43
LNOD	1.223 ± 0.028	1.132 ± 0.021	1.269 ± 0.023	1.086 ± 0.021	1.342 ± 0.031	1.197 ± 0.025	1.104 ± 0.032	1.067 ± 0.027
WNOD	0.517 ± 0.024	0.634 ± 0.021	0.650 ± 0.022	0.501 ± 0.021	0.617 ± 0.035	0.683 ± 0.025	0.417 ± 0.017	0.584 ± 0.030
PROTEIN	5.4 ± 0.06	6.0 ± 0.11	5.9 ± 0.11	5.5 ± 0.07	5.4 ± 0.09	6.5 ± 0.12	5.4 ± 0.09	5.5 ± 0.11
OIL	25.3 ± 0.43	23.8 ± 0.35	25.6 ± 0.37	23.4 ± 0.37	26.6 ± 0.55	24.7 ± 0.42	24.0 ± 0.55	22.8 ± 0.48
STARCH	26.3 ± 0.34	22.2 ± 0.63	25.6 ± 0.27	22.9 ± 0.77	25.6 ± 0.28	25.5 ± 0.46	27.0 ± 0.60	18.4 ± 0.44
C12:0	0.033 ± 0.007	0.027 ± 0.005	0.026 ± 0.008	0.034 ± 0.003	0.030 ± 0.013	0.022 ± 0.010	0.036 ± 0.005	0.032 ± 0.003
C14:0	0.005 ± 0.002	0.025 ± 0.006	0.024 ± 0.006	0.005 ± 0.001	0.005 ± 0.005	0.044 ± 0.009	0.005 ± 0.001	0.006 ± 0.002
C16:0	13.429 ± 0.281	12.019 ± 0.269	11.818 ± 0.233	13.631 ± 0.284	12.630 ± 0.275	11.005 ± 0.280	14.228 ± 0.427	13.034 ± 0.333
C16:1	0.158 ± 0.030	0.068 ± 0.006	0.173 ± 0.028	0.053 ± 0.009	0.247 ± 0.050	0.099 ± 0.005	0.070 ± 0.016	0.037 ± 0.006
C17:0	0.034 ± 0.004	0.042 ± 0.007	0.038 ± 0.007	0.038 ± 0.003	0.030 ± 0.004	0.045 ± 0.014	0.037 ± 0.006	0.039 ± 0.003
C18:0	3.427 ± 0.302	2.008 ± 0.117	1.993 ± 0.163	3.443 ± 0.278	2.065 ± 0.255	1.921 ± 0.209	4.790 ± 0.338	2.096 ± 0.110
C18:1	64.991 ± 0.738	70.011 ± 0.536	69.841 ± 0.575	65.162 ± 0.738	67.243 ± 0.616	72.439 ± 0.515	62.740 ± 1.150	67.583 ± 0.543
C18:1c11	1.186 ± 0.052	1.205 ± 0.048	1.200 ± 0.045	1.192 ± 0.055	1.255 ± 0.063	1.145 ± 0.064	1.118 ± 0.082	1.266 ± 0.071
C18:2	14.933 ± 0.429	13.409 ± 0.327	13.598 ± 0.400	14.744 ± 0.379	15.048 ± 0.534	12.148 ± 0.389	14.817 ± 0.684	14.670 ± 0.347
C18:3	0.332 ± 0.026	0.431 ± 0.137	0.437 ± 0.137	0.326 ± 0.026	0.279 ± 0.018	0.595 ± 0.272	0.386 ± 0.046	0.266 ± 0.015
C20:0	0.623 ± 0.037	0.346 ± 0.022	0.335 ± 0.022	0.634 ± 0.035	0.449 ± 0.024	0.221 ± 0.010	0.797 ± 0.044	0.471 ± 0.017
C20:1	0.340 ± 0.018	0.253 ± 0.021	0.266 ± 0.021	0.327 ± 0.019	0.376 ± 0.017	0.156 ± 0.018	0.304 ± 0.030	0.351 ± 0.023
C22:0	0.211 ± 0.031	0.051 ± 0.008	0.090 ± 0.018	0.172 ± 0.031	0.131 ± 0.032	0.049 ± 0.011	0.292 ± 0.049	0.052 ± 0.012
C22:2	0.147 ± 0.038	0.046 ± 0.008	0.049 ± 0.013	0.143 ± 0.037	0.092 ± 0.022	0.007 ± 0.004	0.202 ± 0.072	0.085 ± 0.010
C24:0	0.640 ± 0.497	0.032 ± 0.004	0.081 ± 0.014	0.591 ± 0.498	0.116 ± 0.025	0.045 ± 0.005	1.163 ± 0.992	0.019 ± 0.005

In the table, the following abbreviations are used: H—the height, NNOD—the number of nodules per plant, MNOD—the weight of nodules per plant, NOD100—the mass of 100 nodules, MPLOT—the weight of nodules per plot, VP—vegetation period, LNOD—the length of the nodule, WNOD—the width of the nodule, C12:0—lauric, C14:0—myristic, C16:0—palmitic, C16:1—palmitoleic, C17:0—heptadecanoic, C18:0—stearic, C18:1—oleic, C18:1c11—vaccenic, C18:2—linoleic, C18:3—linolenic, C20:0—arachidic, C20:1—eicosenic, C22:0—behenic, C22:2—docosadienoic, C24:0—lignoceric acid. 
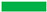
: Highest values of characteristic; 

: High values of characteristic; 

: Low values of characteristic; 

: Lowest values of characteristic.

**Table 2 plants-13-00308-t002:** The value of the coefficients of variation (CV) of the characteristics of chufa in various years of research on the Ekaterinino and Kuban research stations.

Character	2020	2021	ERS	KRS	ERS, 2020	ERS, 2021	KRS, 2020	KRS, 2021
H	9.8	10.3	9.2	10.8	9.3	9.2	10.4	11.2
NNOD	32.5	38.8	39.7	62.4	24.3	33.3	35.0	42.4
MNOD	40.6	36.8	33.4	62.2	20.7	32.2	51.5	40.6
NOD100	23.7	17.9	20.2	20.9	22.2	17.1	22.9	19.0
MPLOT	35.3	48.1	55.1	47.9	29.1	47.2	34.2	49.9
VP	3.0	8.2	12.5	6.8	0.0	0.0	3.4	1.9
LNOD	15.8	12.4	12.1	13.1	11.2	10.1	14.0	12.1
WNOD	31.8	22.0	22.6	28.6	27.1	17.3	19.0	24.7
PROTEIN	8.0	11.6	12.4	8.4	7.9	8.7	8.1	8.8
OIL	11.4	9.5	9.8	10.4	9.9	8.1	10.7	9.5
STARCH	8.6	18.5	7.1	21.7	5.2	8.7	10.4	10.6
C12:0	133.3	122.4	196.6	59.5	193.2	201.6	68.1	47.5
C14:0	308.9	142.9	156.4	136.3	447.2	94.4	121.4	150.1
C16:0	13.3	14.2	12.5	13.2	9.7	11.4	13.4	11.4
C16:1	118.1	59.7	100.9	103.7	90.7	24.0	100.6	77.0
C17:0	69.4	109.9	126.2	55.8	60.2	143.2	74.5	31.8
C18:0	55.8	37.0	51.8	51.1	55.2	48.7	31.6	23.4
C18:1	7.2	4.8	5.2	7.2	4.1	3.2	8.2	3.6
C18:11	27.8	25.2	23.8	29.1	22.6	24.8	32.7	25.1
C18:2	18.2	15.4	18.6	16.3	15.9	14.3	20.6	10.6
C18:3	49.8	201.1	198.1	50.4	29.6	204.3	53.9	25.3
C20:0	37.8	40.7	42.1	34.9	23.7	19.9	24.7	16.2
C20:1	33.0	53.5	51.0	36.7	19.9	53.0	43.6	29.8
C22:0	94.2	100.7	125.1	115.4	108.6	97.5	74.6	105.6
C22:2	166.0	113.4	169.5	164.5	109.1	308.8	160.8	51.0
C24:0	491.2	81.4	107.3	532.9	94.7	52.7	381.4	114.4

In the table, the following abbreviations are used: H—the height, NNOD—the number of nodules per plant, MNOD—the weight of nodules per plant, NOD100—the mass of 100 nodules, MPLOT—the weight of nodules per plot, VP—vegetation period, LNOD—the length of the nodule, WNOD—the width of the nodule, C12:0—lauric, C14:0—myristic, C16:0—palmitic, C16:1—palmitoleic, C17:0—heptadecanoic, C18:0—stearic, C18:1—oleic, C18:1c11—vaccenic, C18:2—linoleic, C18:3—linolenic, C20:0—arachidic, C20:1—eicosenic, C22:0—behenic, C22:2—docosadienoic, C24:0—lignoceric acid.

**Table 3 plants-13-00308-t003:** The significance of the influence of the factors “year”, “station” and their combined interaction on the characteristics of chufa accessions.

Character	Year	Station	Year * Station
H	0.660	0.528	0.212
NNOD	0.000	0.800	0.025
MNOD	0.000	0.131	0.006
NOD100	0.793	0.039	0.101
MPLOT	0.000	0.006	0.004
VP	0.000	0.000	0.000
LNOD	0.002	0.000	0.068
WNOD	0.000	0.000	0.066
PROTEIN	0.000	0.000	0.000
OIL	0.003	0.000	0.489
STARCH	0.000	0.000	0.000
C12:0	0.490	0.357	0.819
C14:0	0.000	0.001	0.000
C16:0	0.000	0.000	0.522
C16:1	0.001	0.000	0.032
C17:0	0.318	0.976	0.413
C18:0	0.000	0.000	0.000
C18:1	0.000	0.000	0.815
C18:1 c11	0.790	0.912	0.071
C18:2	0.004	0.026	0.008
C18:3	0.479	0.425	0.119
C20:0	0.000	0.000	0.072
C20:1	0.000	0.008	0.000
C22:0	0.000	0.008	0.010
C22:2	0.010	0.016	0.682
C24:0	0.224	0.307	0.283

In the table, the following abbreviations are used: H—the height, NNOD—the number of nodules per plant, MNOD—the weight of nodules per plant, NOD100—the mass of 100 nodules, MPLOT—the weight of nodules per plot, VP—vegetation period, LNOD—the length of the nodule, WNOD—the width of the nodule, C12:0—lauric, C14:0—myristic, C16:0—palmitic, C16:1—palmitoleic, C17:0—heptadecanoic, C18:0—stearic, C18:1—oleic, C18:1c11—vaccenic, C18:2—linoleic, C18:3—linolenic, C20:0—arachidic, C20:1—eicosenic, C22:0—behenic, C22:2—docosadienoic, C24:0—lignoceric acid. 

: Significant influence (*p* ≤ 0.05); 

: Close to significant influence (0.05 < *p* < 0.1).

**Table 4 plants-13-00308-t004:** Factor structure of variability according to the complex of morphological, economic and biochemical characteristics.

	Factor 1	Factor 2	Factor 3	Factor 4	Factor 5
H	0.039	0.935	0.049	0.064	−0.046
NNOD	0.907	0.160	0.238	0.017	−0.005
MNOD	0.873	0.149	0.218	0.032	0.186
NOD100	0.020	−0.076	−0.022	0.112	0.945
MPLOT	−0.233	0.117	−0.588	−0.482	0.476
VP	0.338	−0.040	0.836	−0.010	0.044
LNOD	−0.325	−0.341	0.201	−0.561	0.034
WNOD	0.410	0.072	0.461	−0.313	0.488
PROTEIN	−0.049	0.093	0.901	0.079	0.003
OIL	0.006	0.020	−0.174	−0.906	−0.082
STARCH	−0.787	0.322	0.098	−0.257	0.142
Prp. Total	24.1	10.7	20.5	14.1	13.0

In the table, the following abbreviations are used: H—the height, NNOD—the number of nodules per plant, MNOD—the weight of nodules per plant, NOD100—the mass of 100 nodules, MPLOT—the weight of nodules per plot, VP—vegetation period, LNOD—the length of the nodule, WNOD—the width of the nodule, Factor 1—is related to variations in NNOD, MNOD and starch content; Factor 2—H; Factor 3—protein content, VP; Factor 4—oil content; Factor 5—NOD100.

**Table 5 plants-13-00308-t005:** Factor structure of variability in fatty acid content (red marks indicate r > 0.7 (strong correlation with the factor)).

	Factor 1	Factor 2	Factor 3	Factor 4	Factor 5	Factor 6
C12:0	0.362	0.808	−0.050	0.031	0.056	0.002
C14:0	−0.182	0.868	0.140	−0.117	−0.217	0.198
C16:0	0.715	−0.082	−0.371	−0.133	0.261	−0.072
C16:1	−0.083	−0.095	−0.709	0.043	−0.292	−0.172
C17:0	−0.044	0.846	−0.068	0.002	0.160	−0.061
C18:0	0.102	0.163	0.042	−0.061	0.912	0.011
C18:1	−0.748	−0.017	0.273	−0.207	−0.524	−0.114
C18:1c11	−0.102	−0.081	0.087	0.886	−0.044	0.058
C18:2	0.666	−0.056	−0.253	0.395	0.129	0.101
C18:3	0.038	0.078	−0.034	0.036	−0.018	0.824
C20:0	0.372	−0.168	0.017	0.088	0.866	−0.125
C20:1	0.331	0.105	−0.119	0.545	0.132	−0.479
C22:0	0.650	0.188	0.119	0.015	0.157	−0.188
C22:2	0.795	0.000	0.222	−0.142	0.010	0.082
C24:0	0.150	0.147	−0.696	−0.087	0.291	0.276
Prp. Totl	20.2	15.2	9.2	9.0	14.9	7.6

In the table, the following abbreviations are used: C12:0—lauric, C14:0—myristic, C16:0—palmitic, C16:1—palmitoleic, C17:0—heptadecanoic, C18:0—stearic, C18:1—oleic, C18:1c11—vaccenic, C18:2—linoleic, C18:3—linolenic, C20:0—arachidic, C20:1—eicosenic, C22:0—behenic, C22:2—docosadienoic, C24:0—lignoceric acid; Factor 1—C16:0, C18:1, C18:2, C22:0 and C22:2; Factor 2—C12:0, C14:0 and C17:0; Factor 3—C16:1 and C24:0; Factor 4—C18:1c11, C20:1; Factor 5—C18:0, C20:0; Factor 6—C18:3.

**Table 6 plants-13-00308-t006:** The structure of the canonical axes.

	Root 1	Root 2	Root 3
VP	−1.041	0.131	−0.018
STARCH	−0.073	0.692	−0.365
LNOD	−0.060	0.257	0.764
MPLOT	0.117	0.587	0.134
WNOD	0.000	−0.162	0.585
NNOD	−0.173	−0.797	−0.246
PROTEIN	−0.319	0.295	−0.236
OIL	−0.069	0.329	0.240
MNOD	−0.133	0.426	0.311
NOD100	0.070	−0.262	0.251
H	−0.135	0.210	0.146
Cum. Prop	90.3	97.6	100.0

In the table, the following abbreviations are used: H—the height, NNOD—the number of nodules per plant, MNOD—the weight of nodules per plant, NOD100—the mass of 100 nodules, MPLOT—the weight of nodules per plot, VP—vegetation period, LNOD—the length of the nodule, WNOD—the width of the nodule; Root 1—VP; Root 2—starch content and NNOD; Root 3—LNOD and WNOD.

**Table 7 plants-13-00308-t007:** List of *C. esculentus* accessions from the VIR collection used as research materials.

Accession №	VIR Catalog №	Origin
1	1	Russia
2	2	Russia
3	7	Poland
4	8	Bulgaria
5	9	Bulgaria
6	10	Bulgaria
7	11	Mali
8	12	Benin
9	13	Germany
10	14	Ivory Coast
11	15	Ivory Coast
12	16	Ivory Coast
13	17	Ivory Coast
14	19	Ivory Coast
15	20	Ukraine
16	21	Belarus
17	23	France
18	24	Ukraine
19	26	Russia
20	27	Russia

## Data Availability

Data is contained within the article.

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
