# Peer review of "Variability of the Main Economically Valuable Characteristics of Cyperus esculentus L. in Various Ecological and Geographical Conditions"

_plants, 2024, doi:10.3390/plants13020308_

Round 1
Reviewer 1 Report
Comments and Suggestions for Authors
English quality is poor at times. Minor edits are necessary to improve the quality of English.
Author Response
Thank you to the esteemed reviewers for their careful reading of the manuscript and their valuable comments, which have allowed us to improve our work.
Question:
Abstract: Page 1, line 12: Expand VIR.
Answer:
Thanks for the comment. The abbreviation was deciphered. Page 1, line 11
Question:
Abstract: Page 1, line 15: Rephrase the sentence. It is not carrying any meaning.
Answer:
Thanks for the comment. The sentence was rephrased. Page 1, line 18
Question:
Page 4, Line 120: The sentence should read as “significant for these fatty acids” (not “for fatty these acids”).
Answer:
Thanks for the comment. The sentence has been corrected. Page 5, line 160
Question:
Page 7, lines 154 to 156: English is poor. Rephrase the sentence.
Answer:
Thanks for the comment. The sentence has been rephrased. Page 8, lines 208-220
Question:
Discussion: While making comparisons of fatty acid profile, the authors have found significant differences among the plants grown in diverse geographical locations. Such differences could be because of variations in the age of plants, soil conditions, soil nutrient status, soil water status and others which might contribute to such differences have not been taken care of. Therefore, differences are bound to be there. But whether such differences are due to the genotype or due to the environmental conditions or both is not known. The authors may add this information in the discussion part.
Answer:
Thanks to the reviewer for the helpful suggestion. Page 11, lines 321-326
Question:
Tables: The foot notes for tables 1, 2, 3, 4 and 5 may be given, expanding what are NNOD, MNOD, NOD, MPLOT, VP, LNOD, WNOD and others. It is given in Material and Methods, but it is preferred as a foot note since it is easy for the reader to know what is what.
Answer:
Thanks for the comment. Links with abbreviations have been added under each table.
Question:
In Tables 4 and 5, what is meant by factors 1, 2, 3 and so on? That may be given as a foot note.
Answer:
Thanks for the comment. We added links describing the values of each factor.
Question:
Figure: A figure showing the natural habitat of the plant Cyperus may be added.
Answer:
Thanks for the comment. The figure was added.

Reviewer 2 Report
Comments and Suggestions for Authors
The manuscript is in general written in a clear way, easy to read and with appropriate reference to data in tables and figures. The work requires corrections. The purpose of the work should be better articulated.
1. ABSTRACT
- The abstract requires thorough editing. Generaly abstract should contain the most important information about the research being conducted, especially the purpose of the research. Therefore, you should also specify the purpose of the work in this part.
- The abstract should be starting eg. „The study included assessment of chufa collection… in the 2020-2021.”
- The variability 12 of the correlation system of biochemical and yield characteristics over the years of the study – the research included only two years soi t is not enough for this describing.
2. INTRODUCTION
- Please add more information about this crop production in Russia and in other countries not only in Spain.
- The sentences – „The combined application of nitrogen, phosphorus and potassium fertilizers increases the yield [8].” – is too weak.
- Line 46-47 – units,
- research hypothesis should be formulated.
MATERIAL AND METHODS
- this part needs major improvement because it is not clear:
- there is no detailed description of what the research factor was and what specific levels of the factor were used, determining them will make it easier for the authors to formulate a summary;
- Figure 2 – are not legible,
- in the article we can observe lack of information:
-- how many samples were taken for analysis,
-- in what quantities and repetitions,
-- at what time,
-- how it was prepared for analysis,
- abbreviations should be placed under the appropriate tables,
RESUTS
- The information contained in the tables is not understandable 4 i 5 as Factor 1, Factor 2, Factor 3 itp.,
- Similar in table 6.
- There is no abbreviations used in the tables below the tables.
DISSCUSSIN
Sentences should be reconsidered:
- Line 208-212 – „In 2010-2012, a chufa collection was studied in the conditions of Krasnodar region 208 [5]. The investigation of economically valuable characteristics of the chufa collection samples showed that plant height ranged from 49 to 69 cm in 2010-2012 and from 52.5 to 54.5 cm in 2020-2021, tuber weight per plot (1 m2 ) ranged from 101 to 393 g in 2010-2012 and from 180 to 311 g in 2020-2021. - These features were not taken into account in the described studies.
Line 224 – Eyeni at al. - - please add the year of study,
Line 223 - We compared the results of our research with the data obtained by foreign authors.
Summary
The article did not include an economic evaluation! „It can be concluded that there are few articles dedicated to studying the economically 313 valuable characteristics of chufa tubers making the current study relevant.”
Comments on the Quality of English Languagenon
Author Response
Thank you to the esteemed reviewers for their careful reading of the manuscript and their valuable comments, which have allowed us to improve our work.
Question:
The manuscript is in general written in a clear way, easy to read and with appropriate reference to data in tables and figures. The work requires corrections. The purpose of the work should be better articulated.
Answer:
The purpose of the study was revealed more detailed in the introduction. Page 2, line 102-110
- ABSTRACT
Question:
- The abstract requires thorough editing. Generaly abstract should contain the most important information about the research being conducted, especially the purpose of the research. Therefore, you should also specify the purpose of the work in this part.
Answer: The abstract was changed according to the recommendations.
Question:
- The abstract should be starting eg. „The study included assessment of chufa collection… in the 2020-2021.”
Answer: Thanks a lot for the recommendation.
Question:
- The variability 12 of the correlation system of biochemical and yield characteristics over the years of the study – the research included only two years soi t is not enough for this describing.
Answer:
Thank you for your comment, you are right, to identify reliable relationships requires a longer period of study, more than 3 years. But in this case, we can assume trends in the variability of field and biochemical parameters, their dependencies, which can be studied in the subsequent period to confirm their correctness. These trends and the relationships between the indicators are particularly well manifested in contrasting reproduction conditions.
- INTRODUCTION
Question:
- Please add more information about this crop production in Russia and in other countries not only in Spain.
Answer:
Added information according to the comment. Page 2, line 46-48
Question:
- The sentences – „The combined application of nitrogen, phosphorus and potassium fertilizers increases the yield [8].” – is too weak.
Answer:
We changed the sentence in accordance with the comment. Page 2, line 57-60
Question:
- Line 46-47 – units,
Answer:
The units were changed in accordance with the comment. Page 2, Line 67-68
Question:
- research hypothesis should be formulated.
Answer:
We added information to the text of the article according to the comment. Page 3, lines 102-114
MATERIAL AND METHODS
Question:
- this part needs major improvement because it is not clear:
Answer:
Added information to of the section "Materials and methods" according to the comment.
Question:
- there is no detailed description of what the research factor was and what specific levels of the factor were used, determining them will make it easier for the authors to formulate a summary;
Answer:
The main task of the current study was to identify the most promising samples of chufa, yielding a good harvest and maintaining high feed and nutritional quality of nodules in different regions of the country. Such samples can be further used as raw materials to expand the range of food products, including functional, locally produced and to create new high-quality varieties adapted to local growing conditions.
Question:
- Figure 2 – are not legible,
Answer:
We changed Figure 2 and figure title in accordance with the comment.
Question:
- in the article we can observe lack of information:
-- how many samples were taken for analysis,
-- in what quantities and repetitions,
-- at what time,
-- how it was prepared for analysis,
- abbreviations should be placed under the appropriate tables,
Answer:
Lines 374 – 386, 392-398, 400-442, 445-459, 470-471, 486-488 Added information to the text of the article according to the comment: we inserted table 7, supplemented the description of the experiment.
RESULTS
Question:
- The information contained in the tables is not understandable 4 i 5 as Factor 1, Factor 2, Factor 3 itp.,
- Similar in table 6.
Answer: The tables have been corrected in accordance with the comments.
Question:
- There are no abbreviations used in the tables below the tables.
Answer:
Added the abbreviations below the tables.
Comment on the results section.
To represent Figure 1 a flat version was chosen, where only two axes are represented for better clarity of the separation of chufa samples from different places and different years of reproduction.
DISSCUSSION
Sentences should be reconsidered:
Question:
- Line 208-212 – „In 2010-2012, a chufa collection was studied in the conditions of Krasnodar region 208 [5]. The investigation of economically valuable characteristics of the chufa collection samples showed that plant height ranged from 49 to 69 cm in 2010-2012 and from 52.5 to 54.5 cm in 2020-2021, tuber weight per plot (1 m2) ranged from 101 to 393 g in 2010-2012 and from 180 to 311 g in 2020-2021. - These features were not taken into account in the described studies.
Answer:
Thanks for the comment. The text has been corrected in accordance with the recommendations. Page 10, line 280-289.
Question:
Line 224 – Eyeni at al. - - please add the year of study,
Answer:
Thanks for the comment. The year of the study was added. Page 10, line 307.
Question:
Line 223 - We compared the results of our research with the data obtained by foreign authors.
Answer:
Thanks for the comment. The text has been corrected in accordance with the recommendations. Page 9, line 305-308
